# Valorization of Thyme Combined with Phytocannabinoids as Anti-Inflammatory Agents for Skin Diseases

**DOI:** 10.3390/pharmaceutics17101291

**Published:** 2025-10-02

**Authors:** Daniela Hermosilha, Guilherme Trigo, Mariana Coelho, Inês Lehmann, Matteo Melosini, Ana Paula Serro, Catarina Pinto Reis, Maria Manuela Gaspar, Susana Santos

**Affiliations:** 1iMed.ULisboa, Research Institute for Medicines, Faculdade de Farmácia, Universidade de Lisboa, Av. Prof. Gama Pinto, 1649-003 Lisboa, Portugal; danielah@edu.ulisboa.pt (D.H.); mariana.coelho@ff.ulisboa.pt (M.C.); 2R&D&I Department, EXMceuticals Portugal Lda, 1749-016 Lisboa, Portugal; guilherme@tamartech.com (G.T.); ines@tamartech.com (I.L.); matteo@tamartech.com (M.M.); susana@tamartech.com (S.S.); 3Centro de Química Estrutural (CQE), Institute of Molecular Sciences, Instituto Superior Técnico, Universidade de Lisboa, 1049-001 Lisboa, Portugal; anapaula.serro@tecnico.ulisboa.pt

**Keywords:** skin diseases, natural products, anti-inflammatory activity, *Thymus mastichina*

## Abstract

**Background:** Skin diseases of inflammatory origin, such as atopic dermatitis, psoriasis and acne, have a substantial prevalence in the world population. Natural products are particularly important at a topical level. Essential oils are examples of natural products and thyme in particular has been used for medicinal purposes due to its biological properties. **Objectives:** The aim of present work was to study the anti-inflammatory potential of *Thymus mastichina* essential oil, focusing on purified terpene-rich fractions. whose major compounds were thymol and linalool, eucalyptol and α-terpineol, and γ-terpinene and terpinolene, respectively. Additionally, a phytocannabinoid formulation containing cannabidiol (CBD) and cannabigerol (CBG) was evaluated to explore potential synergistic effects. **Methods:** *Thymus mastichina* essential oil was extracted and purified to obtain terpene-enriched fractions, which were used to develop three distinct formulations. These were screened for antioxidant activity using the 2,2-diphenyl-1-picrylhydrazyl (DPPH) assay and assessed for cytotoxicity in HaCaT human keratinocytes. Anti-inflammatory potential was evaluated via gene expression. Selected thyme formulations—alone or in combination with CBD/CBG—were also tested in vivo using a mouse model of acute skin inflammation. **Results:** The antioxidant activity of the three formulations showed a reduction in DPPH radicals. In addition, the formulations demonstrated to be safe in vitro in the human keratinocyte cell model HaCaT. Under PMA-induced inflammatory stress, the fractions modulated-inflammatory gene expression to varying degrees While terpene fractions alone showed moderate activity, their combination with CBD/CBG enhanced the anti-inflammatory response. In vivo, the gel formulations reduced oedema in a mouse model of acute inflammation. **Conclusions:** The data support the safe and effective use of *Thymus mastichina*-derived terpene fractions for topical anti-inflammatory applications. The synergistic effect observed with CBD and CBG suggests that combining essential oil terpenes with phytocannabinoids may offer a novel therapeutic strategy for managing inflammatory skin disorders.

## 1. Introduction

Chronic and acute inflammatory skin diseases—ranging from inflammatory conditions such as atopic dermatitis, psoriasis, and contact dermatitis to more prevalent disorders such as acne, seborrheic dermatitis, and premature skin aging—remain a major challenge globally [1]. These multifactorial disorders often require long-term, sometimes lifelong, management strategies to control inflammation and maintain skin integrity and quality of life. Conventional pharmacological approaches—primarily topical corticosteroids and non-steroidal anti-inflammatory drugs (NSAIDs)—while effective in reducing acute symptoms, carry a substantial risk of side effects upon repeated use, including impaired skin barrier function, microbial resistance, and delayed tissue regeneration [2,3]. The limitations of these standard treatments have boosted increasing scientific interest in safe, multi-targeted alternatives.

Among plant-derived candidates, essential oils (EOs) have gained renewed attention as promising agents due to their diverse bioactivities including anti-inflammatory, antimicrobial, antioxidant, and skin regenerative effects [4,5,6,7]. These effects are largely mediated by terpenes—volatile, lipophilic molecules capable of modulating inflammatory and immune pathways through both direct action and receptor-mediated mechanisms [8,9].

*Thymus mastichina* L. an Iberian endemic thyme species, is particularly rich in bioactive terpenes such as thymol, linalool, eucalyptol, α-terpineol, γ-terpinene, and terpinolene. These compounds have been extensively characterized and associated with a broad range of biological activities, collectively deliver antioxidant, antimicrobial, anti-inflammatory properties [10]. These make it a highly promising botanical candidate for natural cosmetics and dermatological applications, particularly in formulations targeting skin inflammation, oxidative stress, and microbial balance [11,12].

These terpenes, exhibit significant anti-inflammatory effects by targeting multiple molecular pathways involved in the inflammatory response. Several studies have demonstrated that these compounds are capable of inhibiting cyclooxygenase-2 (COX-2) and nuclear factor kappa-light-chain-enhancer of activated B cells (NF-κB), two key regulators of inflammation and immune response in skin pathology [13,14,15].

Nonetheless, the complex and variable nature of crude essential oils (EOs) poses limitations in reproducibility, safety, and regulatory compliance, particularly due to variability in composition depending on geographic origin, plant chemotype, harvesting time, and extraction method. Recent advances in purification and fractionation technologies, such as centrifugal partition chromatography (CPC), facilitate the fractionation of essential oils into terpene-enriched fractions. CPC facilitates the targeted isolation of terpene-rich fractions with enhanced consistency and bioactivity, aligning with the demands of pharmaceutical-grade formulations [16,17].

In parallel, phytocannabinoids such as cannabidiol (CBD) and cannabigerol (CBG) have gained increasing recognition for their anti-inflammatory, antioxidant, and immunomodulatory properties [18].

CBD acts through CB1, CB2, and TRPV1 receptor pathways to attenuate cytokine release (e.g., IL-1β, TNF-α), oxidative stress, and keratinocyte hyperproliferation, while also promoting skin homeostasis [19,20,21,22,23].

CBG, though less studied than CBD, has shown antimicrobial and anti-inflammatory properties, particularly relevant in acne-prone and compromised skin [24,25,26,27].

The “entourage effect” supports the idea that combining terpenes with phytocannabinoids can potentially lead to enhanced therapeutic outcomes compared to using isolated compounds alone [28,29].

This synergy likely results from complementary mechanisms, including enhanced efficacy regarding anti-inflammatory potential, receptor modulation and improved bioavailability. We hypothesized that (i) terpene-enriched fractions of *Thymus mastichina* essential oil would exert measurable anti-inflammatory activity in human keratinocytes and in an acute dermal inflammation model, and that (ii) co-formulation with a CBD/CBG mixture would produce greater effects than either component alone.

In this context, terpene fractions derived from *Thymus mastichina*—rich in anti-inflammatory and antimicrobial compounds—may act synergistically with phytocannabinoids to modulate local immune responses, reduce inflammation and promote skin barrier repair. This rationale supports the development of multi-component, plant-based dermatological formulations tailored to effectively address the multifactorial nature of inflammatory skin disorders.

In this study, we investigate the combined use of CPC-fractionated *Thymus mastichina* EO and a CBD/CBG rich formulation. CPC is particularly advantageous for the purification of cannabinoids and terpenes, as it preserves the integrity of thermolabile and volatile constituents and facilitates the recovery of high-purity fractions suitable for formulations and for in vitro and in vivo testing. This methodology has been successfully applied by our team in previous studies to prepare cannabinoid and terpene formulations for antiviral and anti-inflammatory evaluation [18,27].

Molecular and cellular in vitro assays were performed to evaluate anti-inflammatory activity, and skin regeneration potential. Our work expands the pharmacological relevance of standardized botanical fractions for inflammation-targeted skin applications. Ultimately, the results aim to contribute for the development of novel, multi-targeted phyto-therapeutic dermocosmetic formulations, suitable for regulatory-compliant pharmaceutical, therapeutic and cosmetic applications.

## 2. Materials and Methods

### 2.1. Plant Material

The *Thymus mastichina* subsp. mastichina plant material was supplied by Natural Business Intelligence (NBI) [30]. It was wild-harvested in Palmela, Setúbal, Portugal, during its flowering season (March 2023). After collection, the dried leaves and flowers were separated from the stems for further processing.

### 2.2. Reagents

2,2-Diphenyl-1-picrylhydrazyl (DPPH) and 3-(4,5 dimethylthiazol-2-yl)-2,5-diphenyltetrazolium bromide (MTT), phosphate-buffered saline (PBS), and λ-carrageenan were acquired from Sigma-Aldrich (St. Louis, MO, USA). Antibiotics, culture media, fetal bovine serum (FBS), and phorbol-12-myristate-13-acetate (TPA) were obtained from Thermo Fisher (Whaltham, MA, USA). TaqMan probes were purchased from Alfagene (Carcavelos, Portugal). Primers were acquired from Quantabio (Beverly, MA, USA). TRK lysis buffer was obtained from VWR (Radnor, PA, USA). Carbomer 940^®^ was acquired by Fagron (Barcelona, Spain). All remaining chemicals were of analytical purity grade.

Two commercial formulations were included as positive controls in the in vivo assays to validate the assays: Control 1, a carbopol-based emulsion containing cocoyl caprylocaprate, diethylamine, isopropyl alcohol, ceteth-20, paraffin oil, benzyl benzoate–containing fragrance, propylene glycol, and purified water; Control 2, a botanical/cannabinoid formulation composed of purified water, sunflower seed oil, pentylene glycol, arnica montana extract, glyceryl stearate, PEG-100 stearate, menthol, caprylyl glycol, vanillyl butyl ether, cannabidiol, hemp seed oil, carbomer, hypericum perforatum extract, acrylates/C10-30 alkyl acrylate crosspolymer, sodium hydroxide, n-butyl alcohol, vanillyl alcohol, and vanillin.

### 2.3. Extraction of Thymus mastichina Essential Oil

*Thymus mastichina* essential oil was extracted from the dried leaves and flowers using the hydrodistillation in a Clevenger apparatus, following the procedure outlined in the European Pharmacopoeia [31]. This solvent-free technique efficiently separates volatile from non-volatile component [32]. The hydrodistillation was conducted for 3 h to collect terpene-rich fractions. Extraction yields were calculated according to the following equation:(1)$$\left(\%\right)\;\mathrm{oil}\;\mathrm{content}\;\left(\frac{v}{w}\right)=\;\frac{\mathrm{Volume}\;\mathrm{of}\;\mathrm{oil}\;\mathrm{extracted}\;\left(\mathrm{mL}\right)}{\mathrm{Sample}\;\mathrm{mass}\;\left(g\right)}\;\times \;100$$

### 2.4. Purification of Terpenes and Phytocannabinoids by Centrifugal Partition Chromatography

The purification of terpenes from *Thymus mastichina* essential oil and the isolation of cannabigerol (CBG) were performed using Centrifugal Partition Chromatography (CPC). This technique is especially well-suited for the separation of complex natural matrices, enabling the recovery of bioactive compounds with high purity and minimal thermal or oxidative degradation. For the cannabinoid fraction, CBG was isolated from a *Cannabis sativa* distillate using CPC, while CBD was sourced from BSPG Laboratories [33].

All purification procedures were conducted using a RotaChrom rCPC instrument (RotaChrom Technologies, Purified Solutions, Budapest, Hungary). The biphasic solvent system used in the CPC runs was selected based on preliminary partition coefficient (Kd) determinations for the target compounds. For centrifugal partition chromatography (CPC), we first measured distribution coefficients (Kd = C_stationary/C_mobile phases) for representative terpenes and phytocannabinoids using a standard separatory-funnel (shake–flask) assay at room temperature. We then selected solvent systems that placed target analytes in the Kd ≈ 1–5 range and provided sufficient ΔKd between the target molecules to enable resolution. For terpenes, we used a three-component biphasic family comprising an n-alkane, a nitrile, and a ketone. For cannabinoids, we used an alkane/alcohol/water biphasic system. Under these conditions CBD and CBG exhibited Kd values within 1–5, with ΔKd sufficient to separate them from terpene co-constituents and to resolve cannabinoid-related species Each biphasic system was equilibrated prior to loading to ensure stable stationary-phase retention. Exact volumetric ratios remain proprietary.

The exact solvent composition, volumetric ratios and chromatographic conditions remain proprietary to EXMceuticals Portugal Lda. Fractionation was monitored using UV detection, and collected fractions were subsequently characterized by gas chromatography–mass spectrometry (GC–MS) to verify terpene enrichment, and by high-performance liquid chromatography (HPLC) to quantify and confirm the purity of cannabinoid constituents.

### 2.5. Preparation of Terpene-Enriched Formulations

Following CPC-based fractionation of *Thymus mastichina* essential oil, selected fractions—primarily enriched in bioactive monoterpenes such as 1,8-cineole, linalool, and α-terpineol—were pooled strategically based on their GC–MS profiles and biological relevance. The aim was to concentrate terpene compositions with complementary bioactivities, suitable for therapeutic applications.

These enriched fractions were specifically formulated to support a comprehensive evaluation pipeline, including cell viability and cytotoxicity assays, in vitro anti-inflammatory and antioxidant tests, as well as preliminary in vivo models and molecular analyses focused on skin regeneration and repair.

Each pooled fraction was then formulated into a stable oil-based preparation using medium-chain triglyceride (MCT) oil as the carrier. MCT oil was selected due to its inert nature, favorable skin permeability properties, and regulatory acceptance for both pharmaceutical and cosmetic applications. Three terpene-enriched formulations were prepared and coded as follows: Fraction A (FA), Fraction B (FB), and Fraction C (FC). All formulations were stored in amber glass containers at 4 °C to prevent oxidation

### 2.6. Preparation of Phytocannabinoid-Enriched Formulations

Cann was developed using a proprietary blend of purified CBD and CBG in a specific ratio, previously shown to exert synergistic anti-inflammatory effects in skin models [27]. Building on this, three topical formulations—FA.Cann, FB.Cann, and FC.Cann—were developed, each combining the Cann base with one of the terpene-rich fractions FA, FB, or FC. CBD and CBG were first solubilized in medium-chain triglyceride (MCT) oil, a pharmaceutically approved carrier selected for its high dermal tolerance and efficacy in dissolving lipophilic compounds. To preserve chemical integrity and prevent light-induced degradation, all final formulations were stored in amber glass vials at room temperature, protected from light.

### 2.7. Chemical Analysis by Gas Chromatography-Mass Spectrometry and High-Performance Liquid Chromatography

The terpene composition of thyme essential oil and enriched fractions was analyzed by gas chromatography–mass spectrometry (GC–MS) using a GC-MS-QP-2030 NX system (Shimadzu, Kyoto, Japan) equipped with an HS-20 headspace sampler. Injections were performed with a 1:50 split ratio, using helium as the carrier gas at 1 mL/min. Separation was achieved on a Sapiens-624MS capillary column (30 m × 0.25 mm × 1.4 μm) under a programmed temperature gradient. Mass spectrometric detection was conducted in electron impact mode (70 eV), operating at 200 °C, with acquisition in both SCAN (37–300 *m*/*z*) and SIM modes.

Quantitative analysis of the phytocannabinoids was performed using high-performance liquid chromatography (HPLC) with a Cannabis Analyzer™ for Potency (Shimadzu, Kyoto, Japan) to confirm cannabinoid profile and concentration. These analyses followed standardized internal protocols developed by EXMceuticals Portugal Lda. (Campo Grande, Lisboa, Portugal).

### 2.8. Antioxidant Activity

Antioxidant activity of the fractions of interest was evaluated by the DPPH method, according to a previous study, with some modifications [34]. Briefly, a 40 µg/mL of DPPH solution in methanol was prepared. Methanol was used for the blank. The negative control consisted of adding 180 μL of DPPH solution (40 μg/mL in methanol) and 20 μL of methanol. The positive control (tocopherol) was tested at 0.3% (*v*/*v*). Regarding the samples under study, 180 μL of DPPH solution and 20 μL of each fraction of interest were added. Each reaction mixture was stirred and incubated for a period of 30 min and protected from light, at room temperature. Subsequently, the absorbances (Abs) were read at 532 nm (Cary 60 UV-Vis, Agilent Technologies, Santa Clara, CA, USA). This assay was performed in duplicate, and the antioxidant activity was expressed as percentage of DPPH radical reduction, in accordance with the following equation:(2)$$\left(\%\right)\;\mathrm{DPPH}=\;\frac{{\mathrm{Abs}}_{\mathrm{negative}\;\mathrm{control}\;}-{\;\mathrm{Abs}}_{\mathrm{sample}}}{{\mathrm{Abs}}_{\mathrm{negative}\;\mathrm{control}\;}}\;\times \;100$$

### 2.9. In Vitro Safety in an Immortalized Human Cell Line

Human epidermal keratinocytes (HaCaTs) (CLS 300493) were grown in Dulbecco’s Modified Eagle Medium (DMEM) with high glucose (4500 mg/L), supplemented with 10% (*v*/*v*) fetal bovine serum (FBS), 100 IU/mL of penicillin and 100 μg/mL of streptomycin, and later designated as complete culture medium. Cells were maintained at an atmosphere of 37 °C and 5% of CO_2_ and were trypsinized, every three to four days, when a confluence of approximately 80% was reached.

For the in vitro antiproliferative activity, HaCaT cells were seeded at a concentration of 2 × 10^4^ cells/mL (200 μL per well) in 96-well plates and incubated for 24 h to facilitate their adherence [27]. In the following day, the complete culture medium was removed, and fractions FA, FB, and FC isolated or in combination with Cann at 12.5 μM. Cells incubated with only complete medium, were used as negative control representing 100% of cell viability. Twenty-four hours after incubation, the complete culture medium was removed, cells were washed twice with PBS pH = 7.4 and 50 μL of MTT solution (0.5 mg/mL) in incomplete medium was added in each well. Cells were then incubated at 37 °C with 5% CO_2_ for a period of 2–4 h. After this time, 200 µL of dimethyl sulfoxide (DMSO) was added to solubilize the formazan crystals formed upon reduction of MTT. Subsequently, the absorbance was measured at 570 nm using a BioTek ELx800TM microplate absorbance reader (BioTek Instruments, Inc., Winooski, VT, USA). According to the following equation, the cell viability was determined:(3)$$\mathrm{Cell}\;\mathrm{viability}\;\left(\%\right)=\;\frac{{\mathrm{Abs}}_{\mathrm{sample}}}{{\mathrm{Abs}}_{\mathrm{control}}}\;\times \;100$$
where Abs_sample_ is the absorbance of the sample and Abs_control_ is the absorbance of cells exposed only to complete culture medium, corresponding to 100% cell viability.

### 2.10. Assessment of the Influence of Fractions of Interest in Treatment Condition

The impact on gene expression (by RT-qPCR) in the HaCaT cell line was evaluated for the fractions of interest, FA, FB, and FC, both alone and in combination with Cann under treatment conditions. In this way, phorbol-12-myristate-13-acetate (PMA) was the stress inducing agent used before incubation with the fractions of interest. In previous studies, in 96-well plates different PMA concentrations were tested in the presence of HaCat cells, and 10 µM was selected for testing, through which a cell viability below 20% was attained [27]. Firstly, HaCaT cells were seeded at a concentration of 1.7 × 10^5^ cells/mL (8.5× higher than the concentration used in 96-well plates) (3 mL) in 6-well plates and allowed to adhere at 37 °C under 5 % of CO_2_. After this period, the culture medium was discarded, and cells were incubated with stress inducer, (PMA) at 85 μM (8.5× higher than 10 µM selected in 96-well plates), or complete culture medium (negative control) for 24 h [27]. Following the incubation period, the medium was discarded, and cells were exposed to the isolated and combined fractions of interest at 106.25 µM (8.5× higher than 12.5 µM tested in 96-well plates), or complete culture medium either for the negative control or positive control (cells incubated with PMA). After a 24 h incubation period, the supernatant was discarded and cells were washed with PBS twice to remove cellular debris, HaCaT cells were subsequently collected in pairs, centrifuged at 1500× *g* for 20 min at 4 °C. Then, the pellet was lysed with TRK lysis buffer, and 2-Mercaptoethanol was added to inhibit RNase activity. Two independent assays were performed.

### 2.11. Real-Time Quantitative Polymerase Chain Reaction

Gene expression was quantified using real-time quantitative PCR (RT-qPCR) with TaqMan^®^ probes (Thermo Fisher Scientific, Waltham, MA, USA), a methodology under routine in our lab [27]. Total RNA was extracted and purified from cell cultures using an RNA isolation kit (VWR, PA, USA), following the manufacturer’s protocol. RNA purity was assessed spectrophotometrically, and complementary DNA (cDNA) was synthesized from mRNA in a 20 μL reaction volume. Reverse transcription was performed in a thermocycler (QuantStudio™ 5, Applied Biosystems, Carlsbad, CA, USA) with the following steps: 22 °C for 5 min, 42 °C for 30 min, and 85 °C for 5 min. For qPCR amplification, 5 μL of cDNA was used per reaction well in a final volume of 20 μL, which included 0.4 μL of primer mix (200 nM) and 0.4 μL of probe mix (200 nM). A multiplexing approach was employed to simultaneously assess the expression of 4 gene pairs, each labeled with distinct fluorophores. The amplification protocol consisted of an initial denaturation at 95 °C for 5 min, followed by 45 cycles of 95 °C for 15 s and 58 °C for 1 min. Relative gene expression was calculated using the ∆∆Ct method, with 18S rRNA serving as the internal control. Data were presented as log2-transformed fold changes (log2(2^–∆∆Ct^)) for comparative analysis.

### 2.12. Incorporation of the Fractions of Interest in a Topical Gel

Carbopol^®^ 940 gel was prepared, according to the literature with some modifications [1]. Firstly, the Carbopol polymer was dispersed in water, at a concentration of 0.5% (*w*/*v*), using a mixer IKA Yellowline DI 25 Ultra Turrax, (IKA-Werke GmbH & Co. KG, Staufen im Breisgau, Germany). For the preparation of the gel, NaOH solution was added: 0.4 g of NaOH per g of Carbopol^®^ 940).

The gel formulations were further developed, characterized and used for in vivo testing. FB, FC, FB.Cann, and FC.Cann were incorporated into the gel to evaluate their anti-inflammatory potential in a preliminary carrageenan-induced paw oedema model. A total of five gel formulations were prepared, each containing 1% (*w*/*w*) of a lipid phase: two included 5% (*w*/*w*) of FB and FC; other two included 5% (*w*/*w*) of FB.Cann and FC.Cann, and one contained only CBD/CBG. All final gel preparations underwent additional characterization to confirm their physical and chemical stability.

### 2.13. Characterization of the Unloaded and Loaded Gels

Organoleptic Evaluation

Both the unloaded (base) gel and the bioactive-loaded formulations were assessed for organoleptic properties, including visual appearance (color and homogeneity), texture, odor, and spreadability. These qualitative parameters were evaluated by direct sensory inspection under standardized conditions, in line with established cosmetic formulation assessment guidelines.

pH Determination

To ensure skin compatibility, the pH of each formulation was adjusted to the optimal dermal range (5.0–6.0). pH values were determined at room temperature using a calibrated 744 pH meter (Metrohm, Herisau, Switzerland). Calibration was performed immediately prior to analysis using standard buffer solutions (pH 4.0, 7.0, and 10.0). Measurements were conducted in triplicate to confirm consistency and reproducibility.

Viscosity Measurement

The viscosity of the Placebo Carbopol^®^ 940 gel was measured through the DV-E viscometer (Brookfield Engineering Labs. Inc., Middeborough, MA, USA) and needles n° 3 and n° 4 were used, for a preliminary analysis. This assay was conducted at shear rates from 0.37 to 122.36 s^−1^. The loaded formulations’ viscosity was determined, using the MCR 92 rheometer, which enabled the measurement of small quantities of each gel. The shear rates ranged from 0.10 to 100.00 s^−1^, using a CP50-1 cone plate with 49.961 mm of diameter and 0.00165 rad/Nm of torsional compliance.

Preliminary Stability Assays of Semi-Solid Unloaded and Loaded Formulations containing selected fractions of interest.

The following tests were performed in accordance with the International Guideline ICH Q1A (R2) [35] and previously published literature [36]. Centrifugation stress, as well as heating and cooling assays, were included as part of the preliminary stability assays.

Heating and Cooling

For 7 days, samples were submitted to heat–freeze cycles, consisting of incubation at 25 ± 2 °C in an oven (24 h) and cooling to −5 ± 2 °C in a freezer (24 h). Prior measurements, the frozen samples were allowed to thaw and cool to room temperature. As previously described, pH, viscosity and organoleptic characteristics were assessed after 48, 96, and 168 h.

2.Centrifugation Stress

This assay was adapted from a previously published study [36] with minor modifications. The samples were placed in a water bath at 50 °C (Heidolph MR3001, Heidolph Instruments, Schwabach, Germany) and were consequently centrifuged at 1080× *g* for 30 min (Hermle Z327-K Refrigerated Centrifuge, Hermle Labortechnik GmbH, Wehingen, Germany). All parameters were evaluated at room temperature, before and after centrifugation.

### 2.14. In Vivo Anti-Inflammatory Activity Model

The in vivo efficacy assessment of the formulations was performed using male Wistar rats, supplied from Instituto de Higiene e Medicina Tropical (IHMT) (Lisbon, Portugal). Animals were kept in polypropylene cages at room temperature (20–24 °C) with a relative humidity equivalent to 55 ± 10% and a 12 h cycle of light and darkness. They received standard food and water ad libitum and all experimentation was conducted in accordance with the recommendations of the Animal Welfare Organ (ORBEA) of the Faculty of Pharmacy, Universidade de Lisboa, and approved by the competent national authority, Direção-Geral de Alimentação e Veterinária (DGAV) and in accordance with the EU Directive (2010/63/EU) and all the relevant legislation, including the Portuguese legislation (DL 113/2013, 2880/2015, 260/2016 and 1/2019) for the use and care of animals in research. The anti-inflammatory activity of selected formulations was performed using the carrageenan paw oedema model [1,37]. To carry out this experiment, rats were randomly divided in different groups: negative control, positive controls (two formulations in clinical use—a cream containing a cannabinoid or a diclofenac sodium). The formulations under study were FB, FC, Cann, or combinations of FB or FC with Cann. At least 3 rats per formulation were used. The animals were anesthetized by intraperitoneal injection of a mixture of ketamine at 75 mg/kg and medetomidine at 50 mg/kg. Each formulation under study was applied in the right hind paw of each animal. For the preparation of each formulation, 5% (*w*/*w*) of the respective formulation was used and 1% (*w*/*w*) of a lipid-based absorption promoter was added as a way of reducing the volatility of the essential oil fraction with and without F1.Cann It should be noted that the amount applied of each formulation, as well as the commercial formulations, were similar among all the animals. The negative control was constituted by animals that did not receive any treatment as a way of representing their basal state. Paw oedema was induced by a single subplantar injection of 0.1 mL of λ-carrageenan (1%, *w*/*v*) sterile saline solution, 30 min after topical application of all formulations under study in the right hind paw as well as to negative control animals. The volume of the paws was measured before the topical application of the formulations and carrageenan injection (V_0_) and 4 h later (V_4_), using a plethysmometer (Ugo Basile, Gemonio (VA), Italy). Oedema inhibition was determined according to Equation (4).(4)$$\mathrm{Oedema}\;\mathrm{inhibition}\;\left(\%\right)=\frac{\left|\left(V4-V0\right)\mathrm{negative}\;\mathrm{control}\right|-\left(V4-V0\right)\mathrm{animal}}{\left|\left(V4-V0\right)\mathrm{negative}\;\mathrm{control}\right|}\times 100$$

### 2.15. Statistical Analysis

The in vitro results are presented as mean ± standard deviation (S.D.) from at least two independent experiments, while in the in vivo study, the results are expressed as mean ± standard error of the mean (S.E.M.) of at least 3 animals per group. The statistical analysis was performed using GraphPad Prism 9^®^ (San Diego, SA, USA), with differences were considered significant when *p* < 0.05.

## 3. Results and Discussion

### 3.1. Chemical Analysis of Thymus mastichina Dried Plant

GC-MS analysis of the dried *Thymus mastichina* plant material was conducted to obtain a comprehensive profile of its terpene composition. This analysis enabled the precise quantification of individual compounds, with concentrations expressed in parts per million (ppm), facilitating accurate detection of trace-level constituents (Figure 1). As anticipated, eucalyptol was identified as the predominant terpene in the sample. This critical analytical step helps establish the phytochemical profile of the dried plant, forming the foundation for subsequent experimental stages.

### 3.2. Extraction of Thymus mastichina Essential Oil

The hydro-distillation of dried *Thymus mastichina* plant material resulted in a total essential oil yield of 2.64% (*v*/*w*). This value falls within the range reported in the literature, where hydro-distillation techniques performed over 2 to 4 h typically yield between 0.40% and 6.90% (*v*/*w*) [38]. The oil was subsequently characterized by GC-MS (Figure 2) and subjected to purification aiming to enrich specific terpene fractions for formulation development. The chromatogram analysis is shown in Appendix A.

### 3.3. Purification of Terpenes and Cannabinoids by Centrifugal Partition Chromatography

The purification of terpene fractions from *Thymus mastichina* essential oil and the isolation of CBG from a *Cannabis sativa* distillate were successfully achieved through CPC using a biphasic solvent system optimized based on preliminary partition coefficient studies. The chromatographic runs, conducted with the RotaChrom rCPC system, resulted in well-resolved fractions. Monitoring via UV detection allowed targeted collection of terpene-rich fractions (FA, FB, FC) and highly pure CBG fractions. These were subjected to further analyses (GC-MS and HPLC) to confirm identification and concentration.

### 3.4. GC-MS Analysis of Terpene Fractions

The GC-MS analysis of terpene-enriched fractions revealed distinct compositional profiles. Each chromatogram (Appendix A) demonstrated dominant peaks corresponding to key bioactive compounds. In Appendix A is also presented for the fractions of interest, FA, FB and FC the major terpenes and the respective relative retention indices determined by GC-MS. Table 1 summarizes the two most abundant terpenes identified in each fraction, reporting both their quantitative concentrations (ppm) and their relative abundance (% area).

Analysis of the GC-MS data revealed that each fraction exhibited a predominant terpene: thymol in FA, eucalyptol in FB, and γ-terpinene in FC. These compounds are recognized for their antioxidant and anti-inflammatory properties, supporting the potential of these fractions for dermatological formulations. Although only the top two compounds per fraction are disclosed, minor constituents may contribute to synergistic or modulatory effects on biological activity.

### 3.5. HPLC Quantification of Cannabinoids

HPLC was used to confirm the identification and concentration of the CBG and CBD and their concentration in the formulation Cann. The resulting chromatogram confirmed a purity level exceeding 98%, validating the efficacy of CPC in separating and purifying CBG. Appendix A show representative chromatograms of the pure CBD and CBG, respectively.

A proprietary phytocannabinoid formulation, designated in this paper as Cann, integrates purified CBD and CBG at a defined ratio, previously optimized based on evidence of their synergistic anti-inflammatory effects in a dermatological model.

### 3.6. Development and Characterization of Semi-Solid Gel Formulation

In this study, series of semi-solid gel formulations based on *Thymus mastichina* fractions and cannabinoid mixtures were developed and characterized for their potential application in inflammatory skin conditions. The choice of Carbopol^®^ 940 as the gelling agent was well-justified due to its biocompatibility, non-toxicity, and widespread use in dermatological formulations. The polymer’s rheological behavior was consistent with the existing literature, showing a pseudoplastic (non-Newtonian) profile with decreasing viscosity under increasing shear rates. This shear-thinning behavior is advantageous for topical applications, enabling both spreadability and retention on the skin surface.

The organoleptic and physicochemical analyses confirmed that all formulations were homogeneous, aesthetically acceptable, and maintained an appropriate pH (5–6) compatible with skin physiology, reducing the risk of irritation (Appendix A). Importantly, even upon incorporation of bioactive fractions and cannabinoid mixtures, the pH remained within this desirable range (Appendix A). According to the literature, the pH of the gel becomes more acidic with increasing concentration of Carbopol^®^ 940 due to its acidic properties [39]. In this sense, the pH of the unloaded formulation was adjusted, to reach values between 5 and 6. The skin has a pH between 5 and 6, a buffering capacity to resist alkaline/acid aggression; thus topical formulations, at the pH range mentioned above, should not cause secondary atopic effects [40,41].

The viscosity of the Placebo Carbopol^®^ gel (Appendix A) and that of the formulations tested in the in vivo model of acute inflammation (Appendix A) was also determined. Observing Appendix A, the viscosity of the Placebo Carbopol^®^ gel demonstrated a decrease throughout the shear rate. According to the literature, a rheological study of a liposomal hydrogel based on 0.5% Carbopol was evaluated, in which the formulation demonstrated a non-Newtonian, pseudoplastic behavior [42] corresponding to a decrease in viscosity over the shear rate gradient [43], as expected. In addition, the viscosity range of Carbopol 940 gel is 10× higher than the range reported in the literature corresponding to 40,000–60,000 cP [39]. A possible reason for this difference could be attributed to a different variation in the gelling process. In other words, the addition of the strong base NaOH in this experimental work led to the formation of a three-dimensional network and consequently increased the viscosity of the gel. According to Appendix A, viscosity measurements revealed that the addition of a lipid component, either alone or combined with plant fractions and cannabinoids, resulted in a significant decrease compared to the unloaded formulation (Appendix A). A possible explanation for this difference could be attributed to the low viscosity of the surfactant, which may have resulted in a much lower viscosity in its presence, even in small quantities, having a relevant effect on the final formulation. However, formulations containing Thymus mastichina fractions (FB and FC) and cannabinoid mixtures—individually or combined—preserved their pseudoplastic behavior and showed consistent viscosity profiles (Appendix A). This suggests that the concentrations used did not substantially interfere with the gel matrix structure, which is desirable for maintaining product stability and application consistency.

Furthermore, the preliminary stability studies under heat–freeze cycles and centrifugation stress conditions further supported the robustness of the formulations. Minor variations in viscosity were observed, particularly in the gel containing the FC.Cann, yet all formulations preserved their pH and physical appearance as depicted in Appendix A. These results indicate that the developed systems possess good short-term stability, which is essential for subsequent pharmacological and clinical evaluation.

### 3.7. Assessment of Antioxidant Activity

The antioxidant potential of the terpene-rich fractions, FA, FB, and FC, was evaluated using the DPPH radical scavenging assay, and the percentage of reduction was compared against α-tocopherol, a well-established antioxidant reference. The results are graphically summarized in Figure 3, and the detailed values are presented as mean ± SD. FA exhibited the highest antioxidant activity among the tested samples, with a DPPH radical reduction of 96.58 ± 0.01%, which was virtually equivalent to the activity observed for the positive control, tocopherol. FC demonstrated moderate antioxidant capacity (50.40 ± 0.11%), while FB showed a significantly lower activity, with 9.40 ± 0.58% reduction in DPPH. These results reveal a clear variation in the radical-scavenging capacities of the different fractions, which can be attributed to their distinct chemical profiles, particularly their terpene composition. As reported in the chemical characterization (see Table 1), FA is rich in thymol, a monoterpene phenol known for its hydroxyl-substituted aromatic structure, which enhances its ability to donate hydrogen atoms and stabilize free radicals. This structural feature underpins thymol’s high antioxidant capacity and likely explains the similar performance of FA and tocopherol in this assay [44].

In contrast, the lower activities observed for FC and especially for FB suggest either a lower concentration of antioxidant-active compounds or the predominance of terpenes with weaker hydrogen-donating abilities. FB, which mainly comprised eucalyptol and α-terpineol, demonstrated very low DPPH reduction [10]. Eucalyptol is a bicyclic ether and is a poor H-donor in the DPPH setting; α-terpineol, being a tertiary allylic alcohol also has modest activity compared with phenols. While FC still retained notable activity, from the presence of γ-terpinene and terpinolene, which can donate allylic hydrogens via conjugated diene stabilization, yet are still less potent than phenolic donors. These trends are consistent with prior reports on *T. mastichina* chemotype variability, where thymol- or carvacrol-enriched oils exhibit stronger antioxidant property than eucalyptol-dominant chemotypes [45].

Together, these findings underscore the importance of fractional purification via CPC, as it allows not only for the enrichment of specific bioactive terpenes but also for the functional differentiation of extracts based on their antioxidant profiles. The results support the potential use of FA and FC in topical formulations targeting oxidative stress-related skin conditions.

Several studies have investigated the antioxidant activity of *Thymus mastichina* essential oil using the DPPH assay, yielding varying results. For instance, a study by Oliveira and co-workers [11] reported a poor DPPH radical scavenging activity for essential oils from *Thymus mastichina*. In contrast, Delgado et al. described that essential oils and methanolic extracts from twenty wild Spanish *Thymus mastichina* populations, presented antioxidant properties, with methanolic extracts exhibiting higher DPPH radical scavenging abilities compared to the essential oils [45]. Arantes et al. (2017) reported a high antioxidant capacity of these species, based on harvesting in a different region of Portugal, in Alentejo [46]. These discrepancies in antioxidant activity may be attributed to factors such as geographical variations, environmental conditions, and differences in chemical composition among the plant populations studied. Therefore, while *Thymus mastichina* essential oil has demonstrated antioxidant activity in some studies, the extent of this activity can vary significantly depending on specific conditions and methodologies employed.

### 3.8. In Vitro Safety Evaluation in an Immortalized HaCat Cell Line

The antiproliferative activity of the essential oil fractions FA, FB, and FC was evaluated in the HaCaT cell line, both individually and in combination with cannabinoids (FA.Cann, FB.Cann, and FC.Cann), as shown in Figure 4.

The in vitro assay demonstrated that all essential oils fractions of interest, FA, FB, and FC alone or in combination with Cann, were safe after incubation in the HaCaT cell line at the concentration of 12.5 μM. Although a slight decrease for FA.Cann was observed, the cell viability was superior to 70%. According to ISO 10993-5 [47], none of the formulations showed cell viability below 70%, thus demonstrating its safety.

### 3.9. Assessment of Gene Expression Markers in HaCaT Cell Line

Figure 5 illustrates the transcriptional response of PMA-induced HaCaT cells exposed to terpene-rich essential oil fractions (FA, FB, FC) and their combinations with cannabinoids (FA.Cann, FB.Cann, FC.Cann), focusing on pro-inflammatory, anti-inflammatory, and cannabinoid receptor gene markers. PMA acts as a potent activator of protein kinase C (PKC), which in turn triggers intracellular signaling cascades, particularly the NF-κB pathway. This pathway plays a central role in inflammation by upregulating cytokines such as IL-6, TNF-α, and IL-36G. The experiment aims to determine how these essential oil fractions—alone or combined with CBD and CBG—modulate the expression of key genes related to skin inflammation. Additionally, the PMA-induced stress condition activates the NF-κB pathway, a central transcription factor involved in the regulation of immune and inflammatory responses.

The observed gene expression profiles reflect the modulation of these inflammatory pathways by the tested terpene-rich fractions and cannabinoid formulation, suggesting potential anti-inflammatory or immunomodulatory effects.

#### 3.9.1. Modulation of Pro-Inflammatory and Anti-Inflammatory Cytokines

The pro-inflammatory cytokines IL-6, TNF-α, and IL-36G are central mediators in skin inflammation and were significantly modulated by the treatment with terpene-rich fractions and their combinations with CBD and CBG. IL-6 is a key marker of both acute and chronic skin inflammation, often associated with oxidative stress and skin barrier dysfunction. Its downregulation following treatment indicates an anti-inflammatory effect. Similarly, TNF-α plays a central role in immune activation and is a major contributor to inflammatory skin disorders. The suppression of TNF-α expression suggests a reduction in immune system overactivation, which is beneficial for controlling inflammation. IL-36G, is strongly associated with keratinocyte-driven inflammation and serves as a valuable marker for assessing the therapeutic response of treatments targeting skin inflammation.

Under PMA-induced stress, the gene expression profiles revealed clear distinctions between the effects of individual terpene fractions and their cannabinoid combinations.

PMA stimulation significantly upregulated IL-6, TNF-α, and IL-36G, as expected under inflammatory stress conditions.

Among the terpene fractions, FA led to pronounced upregulation of pro-inflammatory cytokines, suggesting a strong inflammatory stimulus. This profile is consistent with the composition of FA, which is rich in thymol—a monoterpenoid with antioxidant properties, but less effective in suppressing inflammation when used alone. FB triggered moderate increases in IL-6, TNF-α, and IL-36G, while FC was characterized by a moderate IL-6 increase, minor suppression of TNF-α, and modest IL-36G elevation, positioning it as the least pro-inflammatory of the three.

The most significant effect emerged in the phytocannabinoid combinations. The strongest anti-inflammatory effect was observed in FB.Cann, with a marked downregulation of IL-6, TNF-α, and IL-36G, confirming a robust synergistic effect between CBD/CBG and the terpene profile of FB (eucalyptol and α-terpineol). FA.Cann mitigated FA’s pro-inflammatory expression and was close to baseline for all markers, suggesting partial modulation that could be improved, for example, if treatment time was increased. FC.Cann showed mild cytokine suppression with IL-6, TNF-α, and IL-36G, supporting a modest synergistic benefit.

These findings suggest that while the terpene composition of these fractions influences baseline inflammation, their therapeutic potential is better achieved when combined with CBD and CBG.

On the anti-inflammatory side, the focus was on two key anti-inflammatory markers: IL-10, a cytokine that inhibits pro-inflammatory mediators, and PPARγ, a nuclear receptor involved in skin homeostasis and inflammation control. Increased IL-10 expression suggests an enhanced anti-inflammatory response, which is beneficial in conditions characterized by excessive inflammation, such as psoriasis and dermatitis. PPARγ is a key nuclear receptor involved in skin homeostasis, inflammation regulation, and keratinocyte differentiation. Its upregulation is associated with reduced inflammation, enhanced skin barrier function, and protection against oxidative stress.

Under PMA stress, IL-10 and PPARγ were already upregulated as part of the cell’s defense mechanism, consistent with a cellular effort to counteract inflammation.

Treatment with FA, FB, and FC led to a moderate reduction in IL-10 expression, despite partial downregulation of pro-inflammatory markers. This suggests that, at the specific time point assessed by qPCR, the inflammatory response was already in a phase of resolution, reducing the need for sustained IL-10-mediated counter-regulation. A comparable pattern was observed for PPARγ, which remained stable or mildly reduced across individual terpene fractions. In the case of FA, where pro-inflammatory cytokines remained elevated, PPARγ expression was also upregulated, suggesting the activation of a compensatory anti-inflammatory mechanism. This reflects a tightly coordinated regulatory balance between pro- and anti-inflammatory pathways, particularly evident in the modulation of PPARγ during ongoing immune stress. Interestingly, FB.Cann and FC.Cann preserved IL-10 expression closer to the stress-induced level, whereas FA.Cann showed a marked reduction, suggesting differing mechanisms of immune resolution. PPARγ expression revealed a clearer trend. FA maintained the stress-upregulated level, but FA.Cann, FB.Cann, and FC.Cann showed significant downregulation. This indicates that CBD/CBG modulate the PPARγ pathway and may reduce reliance on its action by directly suppressing upstream inflammation. FC.Cann maintained moderate PPARγ levels.

#### 3.9.2. Modulation of Cannabinoid Receptor Genes

GPR55, CNR1, and CNR2 are integral to the endocannabinoid system and are involved in modulating inflammation and immune signaling in skin cells. GPR55, considered a non-classical cannabinoid receptor, is typically associated with pro-inflammatory responses and pain perception, was strongly induced by stress.

CNR1, expressed in keratinocytes, plays a multifaceted role by modulating inflammation, oxidative stress, and wound healing. Dysregulation of CB1 receptor activity has been linked to chronic inflammatory skin conditions. The CB1 receptor also influences pain perception and skin barrier function, highlighting its involvement in both protective and pathological skin processes. CNR2, the gene encoding the CB2 receptor, is primarily expressed in immune cells and skin tissue. CB2 receptor activation exerts strong anti-inflammatory and immunosuppressive effects, helping to control the release of pro-inflammatory cytokines and promoting tissue repair. This receptor is particularly relevant in the context of chronic inflammatory skin diseases, where its activation has been shown to reduce immune overactivation and support the resolution of inflammation.

Among the terpene fractions, FB and FC were more effective than FA in downregulating GPR55. FA treatment resulted in increased expression across all examined cannabinoid receptors, potentially reflecting a transitional immune state. The concurrent upregulation of both pro- and anti-inflammatory markers suggests that, at the time of analysis, the cells were actively modulating inflammation while initiating resolution mechanisms. This pattern highlights the complexity of immune regulation and indicates that Fraction A may be associated with a shift from an inflammatory state towards homeostasis.

Notably, FC.Cann caused the most significant suppression of GPR55 expression, suggesting strong synergistic inhibition. FA.Cann also reduced GPR55 activation but less efficiently than FC.Cann. CNR1 was also upregulated under stress. FC led to its greatest downregulation among the terpene fractions, followed by FB. The combination FB.Cann enhanced this effect, reinforcing the view that FB constituents potentiate cannabinoid action through receptor signaling modulation. FA and FA.Cann had minimal effects on CNR1, indicating that their activity may follow alternative signaling routes. In contrast, CNR2 expression was increased primarily by FA, suggesting a compensatory anti-inflammatory mechanism. FC.Cann showed the strongest capacity to restore and enhance CNR2 expression, confirming its dual activity in both suppressing inflammation and promoting recovery.

### 3.10. In Vivo Anti-Inflammatory Activity Model

Based on the promising in vitro outcomes, essential oil fractions FB and FC were selected for in vivo testing using an acute rat model of inflammation. The carrageenan paw oedema model in rats is considered one of the most widely used in vivo assays to assess the potential anti-inflammatory properties of new formulations, particularly those intended for dermal application [1,37].

In this model, a subplantar injection of freshly prepared 1% λ-carrageenan solution was applied to the right hind paw of male Wistar rats. This procedure was performed 30 min after the topical application of semi-solid gel formulations containing the essential oil fractions FB or FC alone or in combination with cannabinoids FB.Cann and FC.Cann. In addition, the volume of the paw was measured before the beginning of the in vivo assay and 4 h after inflammation induction, thus allowing us to evaluate if the formulations under study were able to reduce the volume of the paw. This experimental protocol is in accordance with the literature [48]. The in vivo results are shown in Figure 6. The negative control presented the lowest inhibition of oedema, close to 0%, confirming the robustness of the inflammation model.

The essential oil fractions FC and particularly FB displayed high anti-inflammatory effect with inhibition of oedema superior to 58 and 177%, respectively.

Notably, when each terpene-rich fraction was combined with the cannabinoid formulation, the anti-inflammatory efficacy was markedly enhanced [49]. The FB.Cann and FC.Cann formulations both demonstrated inhibition levels comparable to those of the positive control groups, highlighting a synergistic interaction between the terpenes and phytocannabinoids. This synergy is likely mediated through complementary mechanisms, including cytokine suppression and cannabinoid receptor regulation.

These in vivo findings support the therapeutic relevance of *Thymus mastichina*-derived terpenes, particularly when combined with CBD and CBG, for the management of acute inflammatory skin conditions. The formulations not only proved safe but also highly effective in reducing inflammation markers at the tissue level, reinforcing their potential in dermatological applications.

Overall, the present work demonstrates that CPC-fractionated *Thymus mastichina* terpene profiles—exert meaningful antioxidant and anti-inflammatory effects both in vitro and in vivo, and that co-formulation with CBD/CBG yields greater anti-inflammatory activity than either component alone.

### 3.11. Mechanistic Considerations and Potential Synergism

Our gel formulations reduced oedema in the acute inflammation model, aligning with established outcomes reported for phytocannabinoids and monoterpene-rich essential-oil constituents in a mouse ear oedema model [48]. The topical administration of a formulation containing cannabinoids was able to reduce ear thickness and inflammatory values in the dermatitis murine model [48].

Emerging CBG data in vivo further supports our data. CBG alleviates atopic dermatitis, reducing epidermal hyperplasia, mast-cell infiltration and inflammatory cytokines while modulating JAK/STAT and NF-κB signaling—mechanisms coherent with our observed gene modulation in keratinocytes [24,50].

In addition, the terpene content from CPC fractions helps to explain efficacy differences. Eucalyptol present in FB typically shows radical-scavenging contributing to anti-inflammatory signaling and may enhance dermal penetration. On the other hand, γ-terpinene/terpinolene present in FC provides intermediate antioxidant/anti-inflammatory activity via NF-κB moderation [51,52].

Although CBD and CBG synergy data in skin are still emerging, the cellular and molecular mechanism do not overlap but instead complement themselves (e.g., CBD: BACH1/Nrf2, TRPV1, PPARγ and CBG: JAK/STAT, NF-κB, TRP family) providing a rationale for combination benefit. Recent CBG in vivo assays in an atopic dermatitis mice model demonstrated a broad cytokine reduction and barrier benefits, complementing CBD’s keratinocyte antioxidant and cellular protection actions observed in vitro, and together they plausibly covered parallel inflammatory routes and oxidative stress [50,53].

Moreover, multi-component cannabis extracts often outperform single cannabinoids against inflammatory cytokines, towards entourage-type interactions. In our formulations, thymol and linalool in FA could boost CBD/CBG effect by reinforcing NF-κB/COX-2 down-modulation and providing high-capacity radical quenching. FB, with eucalyptol despite low DPPH activity, may improve cutaneous penetration of CBD and CBG. γ-terpinene/terpinolene in FC can contribute toward a redox buffering effect that can protect CBD/CBG from oxidative quenching [51,52,54,55].

## 4. Conclusions

This work introduces an innovative approach by combining purified essential oil fractions from thyme with CBD and CBG to modulate key inflammatory and endocannabinoid pathways in keratinocytes, offering a targeted and synergistic strategy for treating inflammatory skin conditions.

The terpene-rich fractions derived from *Thymus mastichina*—FA, FB, and FC—each exhibited a distinct biological profile in PMA-stimulated HaCaT cells. FA, dominated by thymol, triggered a marked inflammatory response consistent with its oxidative potential. FB, rich in eucalyptol and α-terpineol, demonstrated a more balanced anti-inflammatory effect. FC, containing γ-terpinene and terpinolene, showed the mildest pro-inflammatory profile, suggesting a moderate immunomodulatory capacity linked to antioxidant activity.

Importantly, while these fractions alone modulated inflammatory gene expression to varying degrees, their combinations with CBD and CBG significantly enhanced anti-inflammatory outcomes. FB.Cann was particularly effective, showing the greatest downregulation of IL-6, TNF-α, and IL-36G, likely due to synergistic interactions between cannabinoids and oxygenated monoterpenes such as eucalyptol and α-terpineol. FC.Cann provided consistent, though milder, anti-inflammatory effects, maintaining moderate activation of PPARγ and supporting homeostatic recovery. FA.Cann also exhibited strong immunomodulatory benefits, attenuating FA’s pro-inflammatory effects and rebalancing receptor gene profiles. At the gene level, the combinations influenced not only cytokine expression but also endocannabinoid receptors. FA alone upregulated GPR55, CNR1, and CNR2, suggesting an unresolved inflammatory state. In contrast, FB.Cann and FC.Cann effectively suppressed GPR55 and enhanced CNR2 expression, reinforcing their roles in immune modulation and tissue repair.

The in vivo paw oedema model further validated these findings. Both FB and FC reduced inflammation, and their combinations with cannabinoids (FB.Cann and FC.Cann) promoted oedema inhibition levels comparable to standard anti-inflammatory controls. These outcomes correlate with the molecular data, demonstrating that the observed gene modulation translates into measurable therapeutic effects. Hence, the results from the in vivo model reinforce the anti-inflammatory potential of Thymus mastichina-derived essential oil fractions, particularly FB and its combination with CBD/CB (FB.Cann). The significant oedema reduction observed with FB, and the comparable efficacy of FB.Cann and FC.Cann to positive controls, confirm a robust biological effect consistent with the gene expression data obtained in PMA-stimulated HaCaT keratinocytes. The results suggest that targeting cannabinoid receptors alongside terpene-mediated inflammation pathways could provide novel strategies for managing inflammatory skin disorder.

These findings confirm a correlation between molecular modulation and phenotypic outcome, highlighting FB.Cann and FC.Cann as promising synergistic formulations for managing inflammatory skin conditions. Taken together, the evidence supports a dual therapeutic rationale: (1) leveraging terpene profiles to modulate early immune responses and (2) enhancing efficacy through cannabinoid synergy. This phytocomplex-based approach offers a promising avenue for developing natural, multi-targeted topical interventions for inflammatory skin conditions. Further studies are required to validate the formulations’ safety and efficacy in human applications, but these results lay a strong foundation for the continued development of terpene–cannabinoid dermatological therapies.

## Figures and Tables

**Figure 1 pharmaceutics-17-01291-f001:**
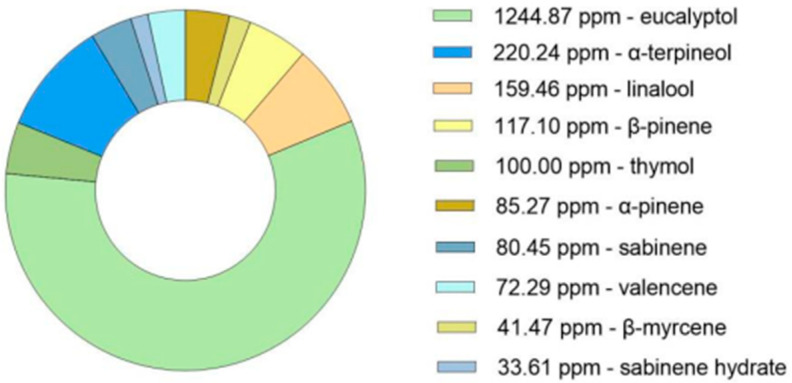
GC-MS analysis (ppm) of the bioactive compounds in the dried plant of *Thymus mastichina*.

**Figure 2 pharmaceutics-17-01291-f002:**
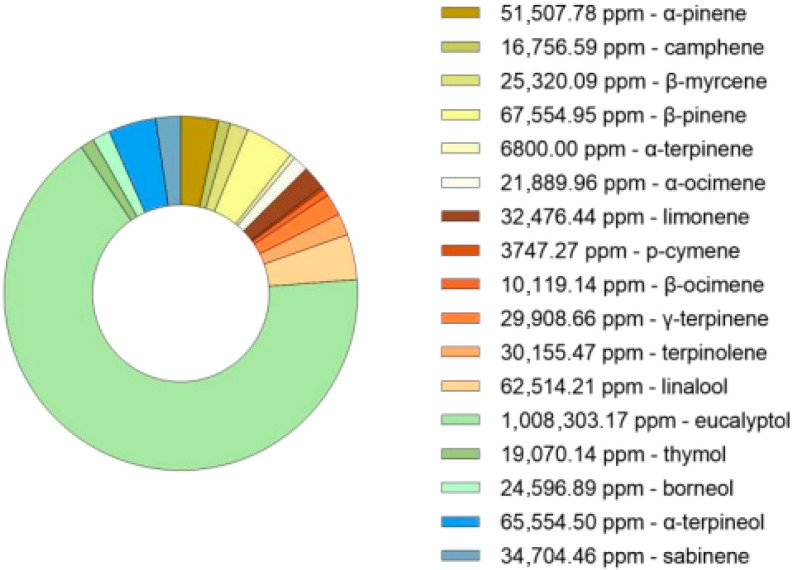
GC-MS analysis (ppm) of the bioactive compounds in the hydrodistillate from *Thymus mastichina*.

**Figure 3 pharmaceutics-17-01291-f003:**
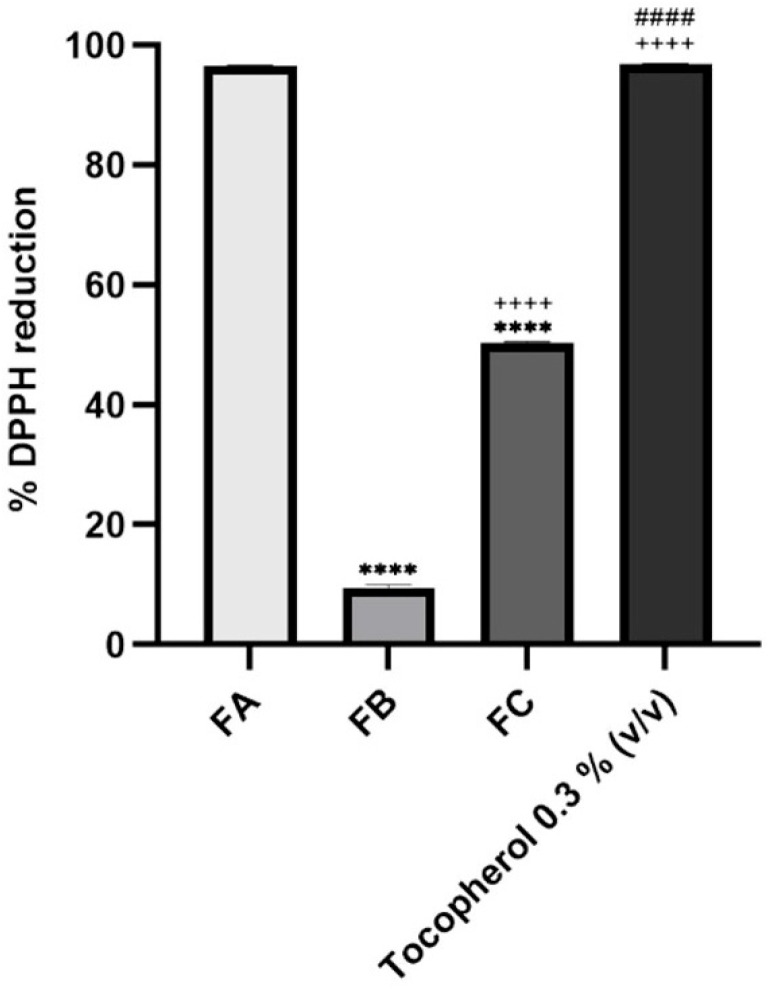
Antioxidant activity of FA, FB, and FC. The positive control, tocopherol, was used at 0.3% (*v*/*v*). The concentration tested for each fraction of interest was 10% (*v*/*v*). Results correspond to mean ± S.D. (*n* = 2). One-way analysis of variance (ANOVA), followed by Tukey’s multiple comparisons test (*p* < 0.05), was used to make comparisons between all the samples. **** *p* < 0.0001 compared to FA; ++++ *p* < 0.0001 compared to FB; #### *p* < 0.0001 compared to FC.

**Figure 4 pharmaceutics-17-01291-f004:**
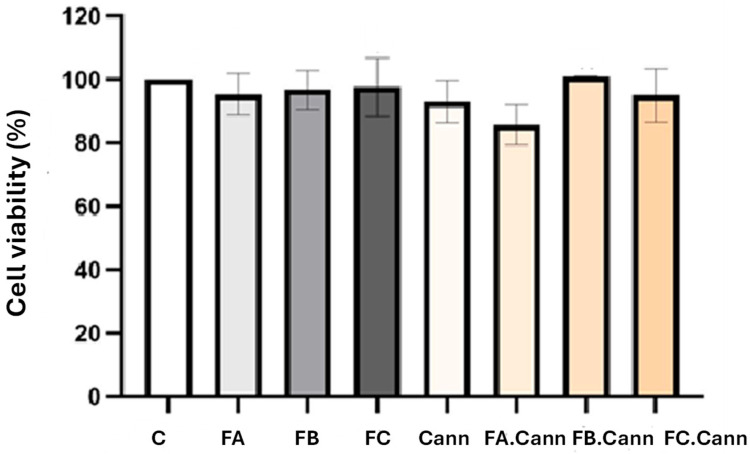
Cell viability (%) of HaCaT cell line by MTT assay 24 h after incubation with MCT oil, FA, FB, and FC alone or in combination with a mixture of cannabinoids (Cann): FA.Cann, FB.Cann, and FC.Cann that were tested at 12.5 μM. Results correspond to mean ± S.D (n = 6). C—HaCat cells incubated with complete medium corresponding to 100% viability.

**Figure 5 pharmaceutics-17-01291-f005:**
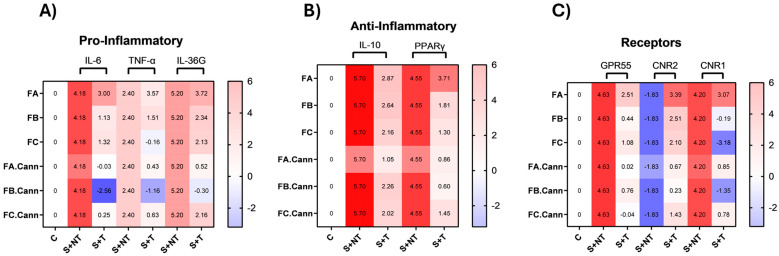
Gene expression heat maps in PMA-induced HaCaT cells. Heat maps illustrating the expression levels of pro-inflammatory (**A**), anti-inflammatory (**B**), and cannabinoid receptor (**C**) genetic markers in HaCaT keratinocyte cells subjected to PMA-induced stress, following treatment with FA, FB, FC, FA.Cann, FB.Can, and FC.Cann. Gene expression changes are shown as average Log_2_(fold change) values (n = 2) relative to the non-stimulated control (**C**), with the following: C: Control group (unstimulated; untreated); S+NT: stressed (PMA-induced); non-treated group; S+T: stressed and treated group. Color scale: Red indicates gene upregulation, while Blue reflects downregulation. Normalization was performed using 18S rRNA as the internal housekeeping gene.

**Figure 6 pharmaceutics-17-01291-f006:**
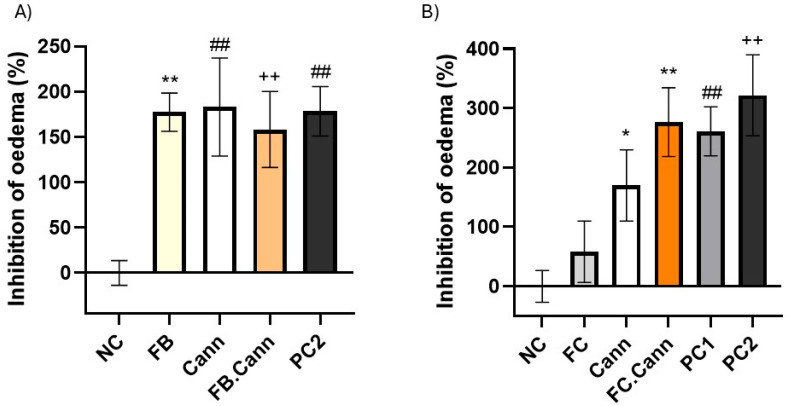
Anti-inflammatory activity of topically applied gel formulations in carrageenan-induced paw oedema rats: (**A**) FB, Cann, FB.Cann and PC2 and (**B**) FC, Cann, FC.Cann, PC1 and PC2. Positive controls: Commercial diclofenac sodium = positive control^1^ and cream with a cannabinoid = positive control^2^. Data corresponds to mean ± S.E.M of three animals per formulation. One-way analysis of variance (ANOVA), followed by Tukey’s multiple comparisons test (*p* < 0.05), was used to make comparisons between all the samples. In (**A**) ** *p* = 0.0072, ^##^ *p* = 0.0057 and ^++^ *p* = 0.0089 in comparison to negative control. In (**B**) * *p* < 0.0332; ** *p* = 0.0053; ^##^ *p* = 0.0084; ^++^ *p* < 0.0021 in comparison to negative control (NC).

**Table 1 pharmaceutics-17-01291-t001:** Quantitative and qualitative analysis of the two major bioactive compounds present in FA, FB, and FC as determined by GC-MS.

Fraction of Interest	Major Terpenes	Quantitative Analysis (ppm)	Qualitative Analysis (% Area)
FA	Thymol	885,805.57	68.39
	Linalool	98,312.59	6.14
FB	Eucalyptol	348,788.01	52.82
	α-Terpineol	467,391.31	13.04
FC	γ-Terpinene	22,080.41	7.08
	Terpinolene	11,177.88	1.94

## Data Availability

The data presented in this study are available on request from the corresponding authors due to commercial reasons. Restrictions apply to the availability of these data. Data were obtained from EXMceuticals Portugal Lda. and are available [from susana@tamartech.com with the permission of EXMceuticals Portugal Lda.

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
