# Peer review of "Valorization of Thyme Combined with Phytocannabinoids as Anti-Inflammatory Agents for Skin Diseases"

_pharmaceutics, 2025, doi:10.3390/pharmaceutics17101291_

Round 1
Reviewer 1 Report
Comments and Suggestions for Authors
This manuscript analyzes the possible impacts of the terpene fractions of Thymus mastichina in conjunction with phytocannabinoids (CBD and CBG) for treating inflammation of the skin. The study is based on a comprehensive approach which includes in vitro (cell culture) and in vivo (animal model) experiments. The original synergistic approach of the study is’most adequate strength and the manuscript is in this respect a suitable submission for the Pharmaceutics Journal. It is clear that the work requires extensive revisions. However, the acceptance of the manuscript is heavily dependent on addressing the issues on methodology, addressing clear problems with the results shown, and fixing many of the the-with the results shown, and fixing many of the manuscript’s remaining terminological inaccuracies.
Introduction: The introduction inflammatory skin diseases, provides a treatment of herbal products. Detailed information is provided on Thymus mastichina and the phytocannabinoids (CBD, CBG). However, the authors could more clearly state the study's hypothesis or research question.
Materials and Methods:
The source and collection time of the plant material are clearly specified (herbarium no?).
The methods of hydrodistillation and Centrifugal Partition Chromatography (CPC) are described. However, the solvent system for the CPC is stated as "proprietary." This significantly impacts the reproducibility of the methodology. The authors must make the methodology more transparent, even without revealing their commercial secrets.
Provide a general solvent class (e.g., "hexane/ethanol/water") without compromising IP.
The coding of the different formulations (FA, FB, FC, and Cann combinations) and the choice of carrier oil (MCT oil) are logical.
The methodology for the DPPH radical scavenging assay is outlined. The procedures for cell viability (HaCaT cells) and gene expression assays are detailed. However, it is mentioned that the 106.25 µM concentration in the gene expression experiment is 8.5 times higher. The rationale for this change in concentration should be more clearly explained.
It is indicated that ethical approvals were secured for the animal studies. The application of the acute edema model (carrageenan-induced) is suitable. Nonetheless, there is a gap in the described methodology concerning the control groups, the commercial products employed, and detailing the respective control groups. The manuscript refers to a commercial product as "a cannabinoid-containing cream," but its exact contents or a standard reference drug (e.g., diclofenac sodium) need to be clarified.
Results and Discussion:
The manuscript presents the results of the GC-MS analysis in ppm. This is important for showing the quantity of the components. However, the text of the manuscript should refer to this data more comprehensively. (Relative retention indices reported in the literature, Relative retention indices (RRI) experimentally determined against n-alkanes should be add)
The in vitro and in vivo results section is present in the manuscript, but the full visuals and data (e.g., tables) are not included in the provided text. A full evaluation of this section is limited without the complete content. Considering findings such as the reduction of DPPH radicals, the safety observed in HaCaT cells, and the reduction in edema, the outcomes seem credible. We noted a reduction in DPPH of 96.58% for FA, which is close to tocopherol, and FB which only reached 9.4%. This sharp contrast requires interpretation. Connect FA and FB results with certain terpenes, for example, thymol’s phenolic structure in FA, eucalyptol in FB, and evaluate with previous research on the variability of T. mastichina.
HaCaT cell viability above 70% (as per ISO 10993-5) is acceptable, but the lack of chronic toxicity studies is a gap (e.g., repeat-dose studies). Discuss this in future directions toward preclinical developmental work.
The discussion section should more deeply analyze the relationship of the results with the existing literature and the potential synergistic effect.
This manuscript requires "Major Revision" for publication in Pharmaceutics. It can be considered for acceptance only after the authors incorporate the following changes:
Making sure all the steps in the methods section are clear, with the exception of proprietary data. Listing all commercial products used as positive controls for in vivo experiments. Providing the rationale for the varying levels of concentration utilized in the gene expression assay. Providing clearer tables and figures to complement the results.
Completion of the above alterations will increase the scientific value and publication acceptability chances of the manuscript.
Author Response
Reviewer 1
This manuscript analyzes the possible impacts of the terpene fractions of Thymus mastichina in conjunction with phytocannabinoids (CBD and CBG) for treating inflammation of the skin. The study is based on a comprehensive approach which includes in vitro (cell culture) and in vivo (animal model) experiments. The original synergistic approach of the study is’most adequate strength and the manuscript is in this respect a suitable submission for the Pharmaceutics Journal. It is clear that the work requires extensive revisions. However, the acceptance of the manuscript is heavily dependent on addressing the issues on methodology, addressing clear problems with the results shown, and fixing many of the the-with the results shown, and fixing many of the manuscript’s remaining terminological inaccuracies.
- Introduction
The introduction inflammatory skin diseases, provides a treatment of herbal products. Detailed information is provided on Thymus mastichina and the phytocannabinoids (CBD, CBG). However, the authors could more clearly state the study's hypothesis or research question.
Reply: Based on reviewer suggestions we have included in introduction section:
“We hypothesized that (i) terpene-enriched fractions of Thymus mastichina essential oil would exert measurable anti-inflammatory activity in human keratinocytes and in an acute dermal inflammation model, and that (ii) co-formulation with a CBD/CBG mixture would produce greater effects than either component alone. “
- Materials and Methods:
The source and collection time of the plant material are clearly specified (herbarium no?).
Reply: We confirm that the source and collection time of the plant material are specified in the manuscript. Information can be seen in Section 2.1 of the manuscript. No herbarium number is available. The sentence was rewritten.
- The methods of hydrodistillation and Centrifugal Partition Chromatography (CPC) are described. However, the solvent system for the CPC is stated as "proprietary." This significantly impacts the reproducibility of the methodology. The authors must make the methodology more transparent, even without revealing their commercial secrets.
Provide a general solvent class (e.g., "hexane/ethanol/water") without compromising IP.
Reply: More details in section 2.4. Purification of terpenes and phytocannabinoids by Centrifugal Partition Chromatography were included in the revised version of the manuscript:
“For centrifugal partition chromatography (CPC), we first measured distribution coefficients (Kd = C_stationary/C_mobile phases) for representative terpenes and phytocannabinoids using a standard separatory-funnel (shake–flask) assay at room temperature. We then selected solvent systems that placed target analytes in the Kd ≈ 1–5 window and provided sufficient ΔKd between the target molecules to enable resolution. For terpenes, we used a three-component biphasic family comprising an n-alkane, a nitrile, and a ketone. For cannabinoids, we used an alkane/alcohol/water biphasic system. Under these conditions CBD and CBG exhibited Kd values within 1–5, with ΔKd sufficient to separate them from terpene co-constituents and to resolve cannabinoid-related species Each biphasic system was equilibrated prior to loading to ensure stable stationary-phase retention. Exact volumetric ratios remain proprietary.”
- The coding of the different formulations (FA, FB, FC, and Cann combinations) and the choice of carrier oil (MCT oil) are logical.
Reply: Thank you.
- The methodology for the DPPH radical scavenging assay is outlined. The procedures for cell viability (HaCaT cells) and gene expression assays are detailed. However, it is mentioned that the 106.25 µM concentration in the gene expression experiment is 8.5 times higher. The rationale for this change in concentration should be more clearly explained.
Reply: The MTT assay was performed in 96 well plates using HaCat at a concentration of 2x104 cells/ml, with the fractions of interest tested at 12.5 µM. For the PCR assays, a higher number of cells was necessary after exposition to the stress inducer PMA. Indeed, the concentration of cells was 8.5-fold higher - 1.7 x 105 cells/mL. To maintain equivalent experimental conditions, the concentration of the test fractions was proportionally increased by the same factor, resulting in a final concentration of 106.25 µM (8.5 × 12.5 µM) in the PCR assays.
- It is indicated that ethical approvals were secured for the animal studies. The application of the acute edema model (carrageenan-induced) is suitable. Nonetheless, there is a gap in the described methodology concerning the control groups, the commercial products employed, and detailing the respective control groups. The manuscript refers to a commercial product as "a cannabinoid-containing cream," but its exact contents or a standard reference drug (e.g., diclofenac sodium) need to be clarified.
Reply: Indeed, as stated in the Methods section, two formulations currently in clinical use were employed as positive controls: a cream containing a cannabinoid and a cream containing diclofenac sodium. The exact commercial names of these products cannot be disclosed in the manuscript due to confidentiality and trademark restrictions. However, their active components, identification, concentrations and therapeutic relevance are fully identified, ensuring transparency and comparability of the results.
- Results and Discussion:
The manuscript presents the results of the GC-MS analysis in ppm. This is important for showing the quantity of the components. However, the text of the manuscript should refer to this data more comprehensively. (Relative retention indices reported in the literature, Relative retention indices (RRI) experimentally determined against n-alkanes should be add).
Reply: We have included Table S1 in supplementary material and for the fractions of interest, FA, FB and FC the major terpenes and the respective relative retention indices determined by GC-MS.
- The in vitro and in vivo results section is present in the manuscript, but the full visuals and data (e.g., tables) are not included in the provided text. A full evaluation of this section is limited without the complete content. Considering findings such as the reduction of DPPH radicals, the safety observed in HaCaT cells, and the reduction in edema, the outcomes seem credible. We noted a reduction in DPPH of 96.58% for FA, which is close to tocopherol, and FB which only reached 9.4%. This sharp contrast requires interpretation. Connect FA and FB results with certain terpenes, for example, thymol’s phenolic structure in FA, eucalyptol in FB, and evaluate with previous research on the variability of T. mastichina.
Reply: In the revised version of the manuscript a more detailed explanation regarding the different antioxidant activity observed for the 3 fractions was included.
“FB is mainly comprised by eucalyptol and α-terpineol demonstrated very low DPPH reduction. Eucalyptol is a bicyclic ether and is a poor H-donor in the DPPH setting; α-terpineol, being a tertiary allylic alcohol also has modest activity compared with phenols. While FC still retained notable activity, from the presence of γ-terpinene and terpinolene, which can donate allylic hydrogens via conjugated diene stabilization, yet are still less potent than phenolic donors. These trends are consistent with prior reports on T. mastichina chemotype variability, where thymol- or carvacrol-enriched oils exhibit stronger antioxidant property than eucalyptol-dominant chemotypes. “
- HaCaT cell viability above 70% (as per ISO 10993-5) is acceptable, but the lack of chronic toxicity studies is a gap (e.g., repeat-dose studies). Discuss this in future directions toward preclinical developmental work.
Reply:
We agree with the reviewer that chronic toxicity and repeat-dose studies are important steps toward the preclinical development of these formulations. The present work, however, was not designed to address long-term safety; its primary objective was to validate the potential of the fractions of interest (FA, FB, and FC) and to explore how their activity could be enhanced in combination with cannabinoids.
Building on these promising results, future studies will focus on documenting safety under repeated exposure. Future planned approaches include repeat-dose testing in reconstructed human epidermis models (7–14 days), histological evaluation of cumulative effects, and in-vitro dermal absorption studies on human skin to estimate systemic exposure and guide dose selection. These will lay the groundwork for subsequent in-vivo studies and, ultimately, controlled clinical trials targeting inflammatory dermatoses such as psoriasis or eczema.
- The discussion section should more deeply analyze the relationship of the results with the existing literature and the potential synergistic effect.
Reply: Aiming to meet reviewer request, in the revised version of the manuscript an additional section was included – “section 3.9. Mechanistic considerations and potential synergism”
“Our gel formulations reduced oedema in the acute inflammation model, aligning with established outcomes reported for phytocannabinoids and monoterpene-rich essential-oil constituents in a mouse ear oedema model [48]. The topical administration of a formulation containing cannabinoids was able to reduce ear thickness and inflammatory values in the dermatitis murine model [48].
Emerging CBG data in vivo further supports our data. CBG alleviates atopic dermatitis, reducing epidermal hyperplasia, mast-cell infiltration and inflammatory cytokines while modulating JAK/STAT and NF-κB signaling—mechanisms coherent with our observed gene modulation in keratinocytes [24,50].
In addition, the terpene content from CPC fractions helps to explain efficacy differences. Eucalyptol present in FB typically shows radical-scavenging contributing to anti-inflammatory signaling and may enhance dermal penetration. On the other hand, γ-terpinene/terpinolene present in FC provides intermediate antioxidant/anti-inflammatory activity via NF-κB moderation [51,52].
Although CBD and CBG synergy data in skin are still emerging, the cellular and molecular mechanism do not overlap but instead complement themselves (E.g - CBD: BACH1/Nrf2, TRPV1, PPARγ and CBG: JAK/STAT, NF-κB, TRP family) providing a rationale for combination benefit. Recent CBG in vivo assays in an atopic dermatitis mice model demonstrated a broad cytokine reduction and barrier benefits, complementing CBD’s keratinocyte antioxidant and cellular protection actions observed in vitro, and together they plausibly covered parallel inflammatory routes and oxidative stress [50,53].
Moreover, multi-component cannabis extracts often outperform single cannabinoids against inflammatory cytokines, towards entourage-type interactions. In our formulations, thymol and linalool in FA could boost CBD/CBG effect by reinforcing NF-κB/COX-2 down-modulation and providing high-capacity radical quenching. FB, with eucalyptol despite low DPPH activity, may improve cutaneous penetration of CBD and CBG. γ-terpinene/terpinolene in FC can contribute for a redox buffering effect that can protect CBD/CBG from oxidative quenching [51,52,54,55].”
This manuscript requires "Major Revision" for publication in Pharmaceutics. It can be considered for acceptance only after the authors incorporate the following changes:
Making sure all the steps in the methods section are clear, with the exception of proprietary data. Listing all commercial products used as positive controls for in vivo experiments. Providing the rationale for the varying levels of concentration utilized in the gene expression assay. Providing clearer tables and figures to complement the results. Completion of the above alterations will increase the scientific value and publication acceptability chances of the manuscript.
Reply: We answered all questions. We expect they meet reviewer expectations.
Reviewer 2 Report
Comments and Suggestions for Authors
This work summarizes the potential of combining thyme-derived terpenes with phytocannabinoids as topical anti-inflammatory agents for skin diseases. By detailing the anti-inflammatory and antioxidant activities of specific terpene fractions from Thymus mastichina, both alone and in combination with cannabinoids, the authors explain the development of novel multi-targeted phyto-therapeutic formulations. The paper highlights the advantages of this synergistic approach, demonstrating through in vitro and in vivo models that the combination significantly enhances the anti-inflammatory response compared to the individual components, offering a promising strategy for managing inflammatory skin disorders. This manuscript has certain research value and can be considered for publication after minor revisions. The following issues can be optimized and modified:
(1)On the Reproducibility of the Centrifugal Partition Chromatography (CPC) Method: The authors state in the Methods section that the CPC solvent system and chromatographic conditions used for separating terpenes and cannabinoids are "proprietary technology" of EXMceuticals Portugal Lda. While understandable for business confidentiality, this significantly compromises the reproducibility of the study. For scientific verification, could the authors provide more information about the type of solvent system?
(2)On the Rationale for Concentration Selection in In Vitro Gene Expression Experiments: In the MTT assay, a concentration of 12.5 μM was used to assess cell safety, but in the subsequent RT-qPCR gene expression experiments, the concentration was increased to 106.25 μM (8.5 times higher). What is the specific reason for this significant increase in concentration? Was its safety to HaCaT cells re-verified at this higher concentration to rule out gene expression changes?
(3)On the Exceptionally High Efficacy of Fraction FB in the In Vivo Anti-inflammatory Model: Based on the data in Figure 6A, when fraction FB was used alone, its inhibition rate on paw edema reached an astonishing more than 177%, which is not only far higher than the positive control drug but also exceeds its combination with cannabinoids (FB.Cann). This data seems to contradict the core hypothesis of the study—synergistic enhancement. Could the authors verify the accuracy of this data point and provide a mechanistic explanation for why a single terpene fraction would exhibit such an abnormally powerful anti-inflammatory effect?
(4)On the Sample Size and Control Group Setup for the In Vivo Experiments: The study mentions that the in vivo animal experiments "used at least 3 rats per group," which is a very small sample size. Could the authors provide a power analysis to demonstrate that n=3 is sufficient to draw statistically significant conclusions?
(5)Furthermore, was a vehicle control group (containing only the gel matrix and MCT oil) included in vivo animal experiments? This is crucial for distinguishing the pharmacological effects of the active ingredients from the effects of the formulation matrix itself.
(6)On the Precise Proportions of Components in the Combination Formulations: The text mentions combining terpene fractions (FA, FB, FC) with the cannabinoid mixture (Cann) but does not explicitly state the concentration or mass ratio of the terpene fraction to Cann in the combination formulations. This information is essential for understanding the dose-response relationship and the extent of synergy. Please provide this information.
Author Response
Reviewer 2
Comments and Suggestions for Authors
This work summarizes the potential of combining thyme-derived terpenes with phytocannabinoids as topical anti-inflammatory agents for skin diseases. By detailing the anti-inflammatory and antioxidant activities of specific terpene fractions from Thymus mastichina, both alone and in combination with cannabinoids, the authors explain the development of novel multi-targeted phyto-therapeutic formulations. The paper highlights the advantages of this synergistic approach, demonstrating through in vitro and in vivo models that the combination significantly enhances the anti-inflammatory response compared to the individual components, offering a promising strategy for managing inflammatory skin disorders. This manuscript has certain research value and can be considered for publication after minor revisions. The following issues can be optimized and modified:
- On the Reproducibility of the Centrifugal Partition Chromatography (CPC) Method: The authors state in the Methods section that the CPC solvent system and chromatographic conditions used for separating terpenes and cannabinoids are "proprietary technology" of EXMceuticals Portugal Lda. While understandable for business confidentiality, this significantly compromises the reproducibility of the study. For scientific verification, could the authors provide more information about the type of solvent system?
Reply: More details in section 2.4. Purification of terpenes and phytocannabinoids by Centrifugal Partition Chromatography were included in the revised version of the manuscript:
“For centrifugal partition chromatography (CPC), we first measured distribution coefficients (Kd = C_stationary/C_mobile phases) for representative terpenes and phytocannabinoids using a standard separatory-funnel (shake–flask) assay at room temperature. We then selected solvent systems that placed target analytes in the Kd ≈ 1–5 window and provided sufficient ΔKd between the target molecules to enable resolution. For terpenes, we used a three-component biphasic family comprising an n-alkane, a nitrile, and a ketone. For cannabinoids, we used an alkane/alcohol/water biphasic system. Under these conditions CBD and CBG exhibited Kd values within 1–5, with ΔKd sufficient to separate them from terpene co-constituents and to resolve cannabinoid-related species Each biphasic system was equilibrated prior to loading to ensure stable stationary-phase retention. Exact volumetric ratios remain proprietary.”
- On the Rationale for Concentration Selection in In Vitro Gene Expression Experiments: In the MTT assay, a concentration of 12.5 μM was used to assess cell safety, but in the subsequent RT-qPCR gene expression experiments, the concentration was increased to 106.25 μM (8.5 times higher). What is the specific reason for this significant increase in concentration? Was its safety to HaCaT cells re-verified at this higher concentration to rule out gene expression changes?
Reply: The MTT assay was performed in 96 well plates using HaCat at a concentration of 2x104 cells/ml. In these assays the fractions of interest were tested at a concentration of 12.5 µM. However, for the PCR assays, after exposition of cells to the stress inducer, PMA, a higher number of cells was necessary. Indeed, the concentration of cells was 8.5 fold higher - 1.7 x 105 cells/mL. To maintain the conditions performed in 96 well plates the concentration of the formulations under study had to be also 8.5 times superior – and so in the PCR assays the fractions of interest were tested at 106.25 μM (8.5 x 12.5 µM).
- On the Exceptionally High Efficacy of Fraction FB in the In Vivo Anti-inflammatory Model: Based on the data in Figure 6A, when fraction FB was used alone, its inhibition rate on paw edema reached an astonishing more than 177%, which is not only far higher than the positive control drug but also exceeds its combination with cannabinoids (FB.Cann). This data seems to contradict the core hypothesis of the study—synergistic enhancement. Could the authors verify the accuracy of this data point and provide a mechanistic explanation for why a single terpene fraction would exhibit such an abnormally powerful anti-inflammatory effect?
Reply: We do not know if we fully understand this comment. Regarding data presented in Figure 6A, the group of animals that received FB presented an inhibition of oedema of 177 ± 30%. However, when compared to the other groups namely - Cann, FB.Cann and PC2 the respective values were 183.10 ± 44.19 %, 195.07 ± 24.40 % and 169.72 ± 36.65 %. Considering the standard deviations, no statistically significant differences were observed between these groups, which is also reflected in the graphical representation.
Importantly, the synergistic enhancement was particularly evident in the combination of FC with Cann (Figure 6B), supporting our main hypothesis. The apparent “exceptional efficacy” of FB alone should thus be interpreted within the variability of the model and not as an anomaly contradicting the overall synergistic effect observed in specific combinations.
- On the Sample Size and Control Group Setup for the In Vivo Experiments: The study mentions that the in vivo animal experiments "used at least 3 rats per group," which is a very small sample size. Could the authors provide a power analysis to demonstrate that n=3 is sufficient to draw statistically significant conclusions?
Reply: We acknowledge the reviewer’s concern regarding the sample size. Nowadays the value of animals in science is becoming increasingly disputed, and the principles of the 3Rs (Replacement, Reduction, Refinement) guided our experimental design. All animal experiments were conducted in strict accordance with national and international regulations on the use of laboratory animals and under ethical approval. Consequently, as a first validation, only the minimum number of animals necessary (n=3 per group) was employed to obtain preliminary, exploratory data on the acute paw edema model.
It is important to emphasize that the main aim of this work was to evaluate the biological potential of the fractions and their combinations. Mechanistic insights were primarily provided by in vitro assays, such as PCR analyses, which complement the in vivo findings while reducing reliance on animal testing. Future studies will include appropriately powered in vivo experiments to confirm these observations.
- Furthermore, was a vehicle control group (containing only the gel matrix and MCT oil) included in vivo animal experiments? This is crucial for distinguishing the pharmacological effects of the active ingredients from the effects of the formulation matrix itself.
Reply: We acknowledge the reviewer’s point regarding the inclusion of a vehicle control. In this study, the focus was on comparing the biological activity of the test fractions to well-established positive controls (a cannabinoid-containing cream and diclofenac sodium). While a vehicle-only group could provide additional information on the contribution of the formulation matrix, it was not included in this exploratory study. However, such control has been considered in our previous study (Pharmaceutics 2020, 12, 1181; doi:10.3390/pharmaceutics12121181). Accordingly, we observed 0.0 ± 0.0 (%) of inhibition with Carbopol 940 gel in an animal model. We agree that this approach can provide additional valuable information, and future work will incorporate vehicle-only groups to better distinguish the effects of the excipients from those of the active ingredients.
- On the Precise Proportions of Components in the Combination Formulations: The text mentions combining terpene fractions (FA, FB, FC) with the cannabinoid mixture (Cann) but does not explicitly state the concentration or mass ratio of the terpene fraction to Cann in the combination formulations. This information is essential for understanding the dose-response relationship and the extent of synergy. Please provide this information.
Reply: The precise concentration and mass ratio of the terpene fractions (FA, FB, FC) to the cannabinoid mixture (Cann) in the combination formulations is confidential and currently considered intellectual property of the development team. This specific formulation detail forms part of a patent application in progress (provisional filing with ref 3435/1.301 E-filed on: August 1, 2025) and cannot be publicly disclosed at this stage. Disclosure is restricted in order to protect the novelty and potential value of the combination formulations, including any unique dose-response or synergy profiles they provide, as required by patent strategy and ongoing IP protection efforts.
Round 2
Reviewer 1 Report
Comments and Suggestions for Authors
I have reviewed the revised version of the manuscript, "Valorization of thyme combined with phytocannabinoids as anti-inflammatory agents for skin diseases," which I had previously reviewed.
I am pleased to see that the author(s) have carefully addressed all of my previous feedback and suggestions and have made the necessary revisions. The changes have significantly improved the quality of the manuscript.
Therefore, I believe that the revised manuscript is now suitable for publication in its current form.
Thank you for your consideration and collaboration.
Sincerely,